# Effect of Opioid Receptor Activation and Blockage on the Progression and Response to Treatment of Head and Neck Squamous Cell Carcinoma

**DOI:** 10.3390/jcm12041277

**Published:** 2023-02-06

**Authors:** Lirit Levi, Elad Hikri, Aron Popovtzer, Avraham Dayan, Amir Levi, Gideon Bachar, Aviram Mizrachi, Hagit Shoffel-Havakuk

**Affiliations:** 1Department of Otorhinolaryngology—Head and Neck Surgery, Rabin Medical Center, Petach Tikva 49100, Israel; 2Translational Research in Head and Neck Cancer, Felsenstein Medical Research Center, Rabin Medical Center, Tel Aviv University, Tel Aviv 6997801, Israel; 3Sackler Faculty of Medicine, Tel Aviv University, Tel Aviv 6997801, Israel; 4Sagol School of Neuroscience, Tel Aviv University, Tel Aviv 6997801, Israel

**Keywords:** head and neck squamous cell carcinoma, opioids, chemotherapy, morphine

## Abstract

Recent studies suggest that opioids have a role in the progression of HNSCC mediated by mu opioid receptors (MOR), however, the effects of their activation or blockage remains unclear. Expression of MOR-1 was explored in seven HNSCC cell lines using Western blotting (WB). XTT cell proliferation and cell migration assays were performed on four selected cell lines (Cal-33, FaDu, HSC-2, and HSC-3), treated with opiate receptor agonist (morphine), antagonist (naloxone), alone and combined with cisplatin. All four selected cell lines display an increased cell proliferation and upregulation of MOR-1 when exposed to morphine. Furthermore, morphine promotes cell migration, while naloxone inhibits it. The effects on cell signaling pathways were analyzed using WB, demonstrating morphine activation of AKT and S6, key proteins in the PI3K/AKT/mTOR axis. A significant synergistic cytotoxic effect between cisplatin and naloxone in all cell lines is observed. In vivo studies of nude mice harboring HSC3 tumor treated with naloxone demonstrate a decrease in tumor volume. The synergistic cytotoxic effect between cisplatin and naloxone is observed in the in vivo studies as well. Our findings suggest that opioids may increase HNSCC cell proliferation via the activation of the PI3K/Akt/mTOR signaling pathway. Moreover, MOR blockage may chemo-sensitize HNSCC to cisplatin.

## 1. Introduction

Pain management is a major concern in cancer patients. Compared with other malignancies, head and neck squamous cell carcinoma (HNSCC) has the highest prevalence of pain, attributed to the rich innervation and function [1]. Over 50% of patients require opioids for pain control [1,2]. Nevertheless, recent findings suggest that opioids may have an influence on disease progression. Retrospective reviews of HNSCC patients correlate opioid use with decreased disease-free survival [1,2]. Additionally, epidemiological studies, including a study performed by our group, found a high prevalence of patients with a history of opioid abuse among patients diagnosed with HNSCC [3,4]. In a different study on supraglottic-SCC, a subtype of HNSCC, we found that patients with a history of opioid abuse present at a younger age and have improved survival, which may suggest a unique disease course [5].

The proposed mechanisms for these epidemiological findings associated opioids with tumor cell proliferation, invasion, angiogenesis, and anti-tumor immune response. Notably, mu opioid receptors (MOR), mediators for opioid effects, were found to be upregulated in several types of cancer cells [6,7,8]. Specifically, our previous study [9] showed an increased expression of MOR on laryngeal-SCC cells compared to normal laryngeal cells. Considering there is an increased expression of MOR over HNSCC tumor cells and the common use of opioids among HNSCC patients, the role of MOR in those patients should be investigated. Although there is some evidence that opioids may contribute to HNSCC development and progression, the effects of activation or blockage of MOR in tumor cells is unclear. 

PI3K/AKT/mTOR signaling axis, a frequently activated signaling route in HNSCC [10], promotes cell survival and growth and is the most frequently mutated pathway in HNSCC [11,12]. This pathway is the objective of targeted therapy and has a role in cell resistance to chemotherapy [13]. Key players in these pathways include EGFR, PI3K, Akt, mTOR, and PTEN [11], which are commonly mutated in lung cancer, and are found to be upregulated by morphine via MOR in non-small-cell lung cancer [7,14].

This study aims to investigate, by means of in vitro and in vivo studies, the effects of MOR activation and blockage on HNSCC, with relation to disease progression, response to therapy, and activation of the PI3K/AKT/mTOR signaling axis. We hypothesize that these effects can be measured in vitro by parameters such as cell proliferation and migration, as well as in vivo by tumor volume. The findings of this study might lay grounds for determining the impact of opioid use in HNSCC patients. 

## 2. Methods

### 2.1. Experimental Design and Setting

The study was performed at the Center for Translational Research in Head and Neck Cancer, Felsenstein Medical Research Center, Rabin Medical Center. The study investigated the in vitro and in vivo behavior of commercial human HNSCC cell lines when opioid agonists or antagonists are added, with or without conditions that correspond with therapy for HNSCC. Each in vitro assay was repeated at least three times and analyzed statistically by Student’s *t*-test. 

### 2.2. Cell Cultures and Reagents

Human HNSCC cell lines utilized in this study included: Cal-33 (tongue SCC, Cellosaurus, accession number; CVCL_1108), FaDu (hypopharynx SCC, ATCC, accession number; HTB_43), HSC2 (oral cavity SCC, Cellosaurus, accession number; CVCL_1287), HSC3 (oral cavity SCC, Cellosaurus, accession number; CVCL_1288), KYSE 70 (esophageal SCC, Cellosaurus, accession number; CVCL_1356), KYSE 180 (esophageal SCC, Cellosaurus, accession number; CVCL_1349), and LB771 (head and neck SCC, Cellosaurus, accession number; CVCL_1369). All cell lines were maintained in a humidified atmosphere of 5% CO_2_ in air at 37 °C incubator in the appropriate cell culture media that was supplemented with 10% heat-inactivated fetal calf serum, 2 mML-glutamine, penicillin (100 U/mL), and streptomycin (100 μg/mL).

### 2.3. Interventions 

Four selected cells were treated with cisplatin (ABIPLATIN 1 mg/mL, TEVA, Haarlem, Amsterdam), naloxone (Naloxona 0.4 mg/mL, Kern Pharma, Terrassa, Spain), and morphine (Morfina hidrohlorids 10 mg/mL, Kalceks, Riga, Latvia) in a concentration gradient to determine the cytotoxicity and sensitivity doses to each agent. After establishing the concentrations required for treatment, cells were incubated with cisplatin (40 µM, 25 µM, 20 µM) with or without 10 µM naloxone or 10 nM morphine and compared with controls (saline, naloxone, or morphine).

### 2.4. Western Blotting

For immunoblotting, cellular and tissue homogenates were incubated with lysis buffer run on SDS-PAGE in polyacrylamide gel, transferred onto a nitrocellulose membrane, and developed with specific primary and secondary antibodies. The primary antibodies used in this study were anti MOR-1 and anti-phosphorylated (activated) form of AKT (p-AKT) and S6 (p-S6). Visualization of immunoreactive bands was achieved using fluorescent detection. Quantification of immunoblotting was obtained by densitometric analysis using ImageJ Software.

### 2.5. XTT Cell Proliferation Assay

For measuring in vitro cell growth, cells (5000 cells/well) were seeded into 96-well plates to a final volume of 100 μL and were allowed to grow for 24 h with the appropriate treatments of morphine or naloxone with and without combinations of cisplatin versus control. Subsequently, the activated XTT reaction solution was added to each well for additional 2–4 h incubation at 37 °C. Finally, the reaction was measured at a wavelength of 450 nm against a background control by a microplate reader.

### 2.6. Migration Assay

Measurement of in vitro cell migration was performed in a twenty-four transwell units with 8 μm pore size. Cells pretreated with morphine or naloxone treatments were incubated with 0.2 mL serum-free media in the upper chamber and allowed to migrate to media with 20% serum to the lower chamber through the pores for 18 h. Cells attached to both sides of the membrane were washed twice with PBS. Cells on the upper side of the membrane were removed by cotton swabs, and the migrating cells at the bottom of the membrane were visualized using a microscope, photographed, and counted.

### 2.7. Animal Model Hosting Human HNSCC

Male athymic nude mice were used for this study (8 weeks old and 27–38 g). Animals used in this study were humanely cared for, and all experiments were performed under the aegis of a protocol approved by the Tel Aviv University and were in compliance with the Guide for the Care and Use of Laboratory Animals, National Research Council. A total of 100 µL of 2 × 10^6^ HSC-3 HNSCC cells (derived from the in-vitro-used commercial HNSCC cell lines) were subcutaneously injected to the flank of the mouse. Tumor volume was measured twice a week with a caliper. The tumor volume was calculated using the formula: 1/2 × length (mm) × [width (mm)]^2^.

### 2.8. Interventions and Medications

In order to test the effect of MOR blockage on tumor’s growth, structure, and surrounding, the studied mice were administered with MOR antagonists in addition to or without an oncological intervention.

When the tumors reached an average volume of 100 mm^3^, the relevant subgroups received treatment as follows. Thirty animals were randomly divided into 4 groups: (1) control, (2) naloxone (1 mg/kg once a day), (3) cisplatin (3 mg/Kg every two days), (4) cisplatin (3 mg/Kg every two days) and naloxone (1 mg/kg once a day); different drugs were administered by separate injections. The selected doses for naloxone and cisplatin administration were based on data and findings of previous research [15,16,17,18]. The drugs were diluted in a 0.9% saline solution and administered in a 100 µL via intraperitoneal injection. Experiments were continued to monitor and measure the tumors for 6 weeks or up to 1500 mm^3^. Tumor change in size were compared between the different study groups.

### 2.9. Statistical Analysis

SPSS version 23 software (IBM, Armonk, NY, USA) and GraphPad Prism version 6.0 (Graphpad, La Jolla, CA, USA) was used to analyze and plot the data. Data were presented as mean +/− standard error (SE). Differences between two groups were analyzed using the independent *t*-test, whereas the analysis of variance (ANOVA) test was used to compare multiple groups. The Bonferroni correction was applied for multiple comparisons. Linear model fit significance was calculated using F-test. Differences in the proportions of observations of two categorical variables were tested with a G-test.

## 3. Results

### 3.1. MOR-1 Expression in HNSCC Cell Lines

All seven HNSCC cell lines examined express MOR-1 in Western blotting analysis (WB), as shown in Figure 1. While Cal-33, FaDu, and HSC-2 cell lines demonstrate relatively higher expression of MOR-1, HSC-3 shows modest expression. We, therefore, focused on these four cell lines for additional analyses, investigating the effects of morphine and naloxone on cell lines with the richest and modest levels of MOR-1.

### 3.2. MOR Agonist and Antagonist Effects on Cell Proliferation

First, cell proliferation and toxic doses of morphine were assessed in a dose-dependent fashion with gradually increasing dosages (Appendix A); the morphine dose used was chosen accordingly. The addition of morphine (10 nM) induces an increase in cell proliferation in all four HNSCC cell lines examined (Figure 2). This increase is statistically significant in two cell lines. Cell viability increases by 13% (*p*-value < 0.05) and 24% (*p*-value < 0.05) in the FaDu and HSC2 cell lines, respectively (Figure 2b,c).

Next, cell proliferation and toxic doses of naloxone were assessed in a dose-dependent fashion, with gradually increasing dosages of 0.1 µM, 1 µM, 10 µM, 20 µM, and 40 µM. The toxic effect of naloxone is observed in concentrations over 20 µM. (Appendix A). A non-toxic dosage of naloxone (10 µM) induces an inconsistent effect on cell proliferation (Figure 2). A non-significant reduction of 4% in cell viability is demonstrated in HSC-3 (Figure 2d). A significant increase in cell proliferation by 19% is observed in the HSC-2 cell line (*p*-value < 0.05), while a mild non-significant increase is observed in Cal-33 and FaDu cell lines, by 13% and 4%, respectively (Figure 2a,c).

### 3.3. MOR Agonist and Antagonist Effects on MOR-1 Expression

Densitometric analysis of WB demonstrates a trend of increased MOR-1 expression in all cell lines with the addition of morphine (10 nM, Figure 3b).

### 3.4. MOR Agonist and Antagonist Effects on Downstream PI3K/AKT/mTOR Signaling Axis

Densitometric analysis of WB demonstrates that the addition of either naloxone (10 µM) or morphine (10 nM) alters phosphorylation of AKT and P6 in all cell lines (Figure 3c,d). Levels of phosphorylated-AKT (p-AKT) increase with the addition of morphine in all four cancer cell lines, significantly in HSC-2 (*p*-value < 0.01). Levels of p-AKT are attenuated with the addition of naloxone in two out of four cell lines and significantly increase in the HSC-2 cell line (*p*-value < 0.01, Figure 3b). A trend of increasing levels of phosphorylated-S6 (p-S6) with the addition of morphine is observed in all four cancer cell lines (Figure 3c). A significant increase in p-S6 is observed with the addition of naloxone (*p*-value < 0.01) in HSC-2 cell lines.

### 3.5. MOR Agonist and Antagonist Effects on Response to Cisplatin

Cisplatin induces a reduction in cell viability in a dose-dependent manner (Appendix A). HNSCC cell lines were incubated with various concentrations of cisplatin together with increasing doses of naloxone (Appendix A). The addition of naloxone (10 µM) to cisplatin (20 µM) results in a decline in cell viability in all cell lines. A reduction of 20% for HSC2, 17% for FaDu, and 8% for Cal-33, and a significant decline by 46% (*p*-value < 0.01) for HSC-3 (Figure 4a–d), suggesting a synergic cytotoxic effect between cisplatin and naloxone. The addition of morphine (10 nM) to cisplatin does not result in any consistent or significant effect on cell viability. To determine whether a synergetic effect occurred, we performed an isobolographic analysis of naloxone and cisplatin (Figure 5). ED50% values for naloxone (Figure 5a) and cisplatin (Figure 5b) were interpolated from the data using piece-wise cubic spline interpolation. The potency ratio at 50% effect R50 = 1.16, is not consistently different from the potency ratio at maximum effect R100 = 1.07 (*p* = 0.74, g-test). Therefore, the isobole of additivity can be viewed as linear [19]. The combination of naloxone (10 µM) and cisplatin (20 µM) falls below the isobole but within the 95% confidence interval, thus, a synergetic effect cannot be concluded (Figure 5c). However, the total effect of the combination of naloxone (10 µM) and cisplatin (20 µM) is 84.6% +/− (7.8%), consistently higher than 50% effect (*p* = 0.02, *t*-test). Therefore, we performed an isobolographic analysis using an effect level higher than 50% [19], specially and effect level of 84% (Figure 5d). The combined dose of naloxone and cisplatin falls well outside the 95% confidence interval, indicating a synergistic effect.

Cell migration assays were performed for all four cell lines (Figure 6). The addition of morphine (10 nM) significantly promotes cell migration in HSC-3 by three fold (*p*-value < 0.01, Figure 6e), while it significantly reduces migration in Cal-33 by 22% (*p*-value < 0.05, Figure 6b). The addition of naloxone (10 µM) shows an inhibitory cell migration trend for FaDu by 30% (Figure 6c).

### 3.6. In Vivo Administration of MOR Agonist and Antagonist and Their Effect on Chemotherapy

We used a mice model injected with HSC-3 tumor cell lines to further establish the proliferative effects MOR agonist and antagonist that were observed in the in vitro assays. Administration of naloxone (1 mg/Kg) decreases tumor proliferation by 37% compared to control. Analyzing the outcome in HSC-3-injected mice treated with a combination of cisplatin and naloxone, it demonstrates a synergistic effect from the addition of naloxone results in increased tumor volume reduction by 64% compared to cisplatin alone (*p*-value < 0.01, Figure 7).

## 4. Discussion

Our study clearly demonstrates the presence and function of MOR in HNSCC on pre-clinical models. We found that exposure to morphine results in tumor cell proliferation and migration, accompanied by phosphorylation of AKT and S6, indicating the activation of the PI3K/AKT/mTOR signaling pathway. Furthermore, by combining cisplatin with naloxone, we were able to demonstrate a significant synergistic cytotoxic effect between these two agents both in vitro and in vivo.

The role of opioids in cancer cell proliferation, invasion, metastasis, and angiogenesis promotion was previously described in non-small-cell lung cancer (NSCLC), breast cancer, nasopharyngeal carcinoma, renal cell carcinoma, and hepatocellular carcinoma [7,14,15,16,18,20,21,22,23,24]. This role was supported by epidemiologic reports as well [1,2,25,26,27,28,29]. Our findings correlate between exposure to opioids and tumor proliferation and progression. Gorur et al. [30] recently studied the effect of DAMGO ([D-Ala^2^,N-Me-Phe^4^,Gly^5^-ol]]-enkephalin), a MOR-selective agonist on HNSCC cell lines. MOR activation by DAMGO increases proliferation, invasion, and migration, compatible with our results, while methylnaltrexone, a DAMGO antagonist combined with DAMGO, has the opposite effect. Interestingly, our results show that naloxone, an opioid antagonist, has an independent negative effect on cell proliferation. Moreover, we were able to demonstrate a synergistic effect between this opioid antagonist and chemotherapy. These are all novel findings in HNSCC research that warrant further investigations.

The opioid receptors are divided into three major subgroups, µ, κ, and δ, and are all members of the G-protein-coupled receptor superfamily. MOR (µ) is the main target for opiates [31,32]. All our HNSCC cell lines express MOR-1 to some degree. However, there is no correlation between the degree of MOR-1 expression and the effects of morphine or naloxone on proliferation and downstream signaling. This discrepancy can be explained by other opioid receptor subtype or oligomerization, or by the fact that morphine or naloxone activity on cells may be mediated by these nonclassical opioid receptors [18]. An alternative route of the naloxone inhibitory effect on HNSCC cell proliferation was suggested by previous studies investigating the role of opioid growth factor (OGF), an endogenous native opioid that interacts with the OGF receptor (OGFr) to inhibit cell proliferation [33]. These studies show that when cells are exposed to low naltrexone (opioid antagonist) doses, the OGF–OGFr complex is upregulated and has an inhibitory effect on cell proliferation [34]. Other opioids, such as morphine, do not affect the OGF–OGFr complex [35,36], hence, this mechanism cannot explain the effect of morphine on cell proliferation and the PI3K/AKT signaling pathway.

Another receptor that might take part in the opioid agonists and antagonists effect on cancer cell proliferation, migration, and invasion is the epidermal growth factor receptor (EGFR, also known as erbB-1) [14]. EGFR is a receptor tyrosine kinase (RTK), which has been shown to correlate with poor outcomes when overexpressed in HNSCC [37,38]. Several therapies targeting EGFR, such as cetuximab and multi-tyrosine kinase inhibitors (TKI), are utilized in advanced HNSCC. However, overall survival of these patients remains low [39,40,41]. The interaction between morphine and EGFR-targeted therapy was previously described in other cancer types [14], suggesting that morphine may induce a resistance effect to TKIs. Further investigation is required to understand the influence of opioids therapy on anti-EGFR-targeted therapy in HNSCC.

Regarding signaling pathways, our findings suggest that morphine may have a role in the activation of the PI3K/AKT/mTOR axis, a dominant signaling pathway in HNSCC as well as in other cancer types [7,10,14,18,31,42]. This pathway has an essential role in promoting tumor growth, invasion, and metastasis. AKT, a serine/threonine-specific protein kinase, can induce cellular survival by inhibition of apoptosis and protein synthesis. AKT integrates signals from growth factors to activate mTOR, which promotes a signaling cascade. Ribosomal protein S6 is a main downstream protein in the cascade [42,43]. Lennon et al. investigated the role of MOR overexpression in NSCLC [31], and their results indicate that MOR stimulates cell progression via EGFR-dependent PI3K/AKT/mTOR signaling pathway activation [7,31]. Co-activation of EGFR via opioids receptors was supported by others as well [14].

Another downstream pathway, investigated in both HNSCC and NSCLC [7,44], is STAT3. The role of STAT3 was confirmed to be regulated by both EGFR and MOR in NSLCL [7], which could be an interest for further investigation, concerning the role of opioid agonists and antagonist in HNSCC.

The synergistic effect of opioid antagonists and targeted chemotherapy/anti-cancer therapies was previously discussed in other types of cancer. Singleton et al. investigated the synergistic effect of methylnaltrexone, a selective peripheral μ-type opioid receptor antagonist, with anti VEGF (Avastin) and 5FU in endothelial cells, inhibiting VEGF-induced angiogenesis [24,45]. This was also observed by others [15,18]. In our study, a synergic cytotoxic effect was observed when combining cisplatin with nontoxic dose of naloxone, suggesting that MOR blockage has a chemo-sensitizing effect in HNSCC cells [17]. Previous in vivo and in vitro studies on NSCLC reported inhibition of PI3 kinase and Akt phosphorylation in response to MOR antagonists, with subsequent attenuation of cell proliferation invasion and migration [7,14,18,20]. Furthermore, inhibition of the PI3K/AKT signaling pathway decreases cancer cell resistance to cisplatin [13,46,47]. A similar result was observed in breast cancer cells, via the MAPK/ERK signaling pathway [22].

Rapamycin, an mTOR inhibitor, induces apoptosis and inhibition of cell growth [48]. The synergism between mTOR inhibitors and cisplatin was previously demonstrated on several different cancer cell types, including head and neck cancer [47,49,50,51,52,53]. The proposed mechanism for the synergistic effects includes a downstream regulatory mTOR-mediated pathway, shifting the cell balance towards apoptosis and enhancing the effects of cisplatin [53]. Our finding aligns with those of previous studies and support the inhibitory effect of naloxone via the PI3K/AKT/mTOR signaling pathway.

These findings suggest the importance of opioids in cancer cell resistance to chemotherapy, and the potential of opioid antagonists in sensitizing HNSCC to targeted therapy, such as chemotherapy or even radiation. Therefore, our findings support what was previously suggested by several authors in other cancers, that opioid antagonists could be utilized as adjuvant therapy in advanced stage cancers [17,20,24,31].

One limitation of our study is that naloxone, the non-selective opioid antagonist used, although it has the highest affinity towards MOR, may also target other opioid receptors such as the delta (DOR) and kappa (KOR) receptors in addition to the mu (MOR) receptor. This means that the effects of naloxone on cancer cell progression may not be solely due to its interaction with the MOR receptor. To gain a more specific understanding of the role of the MOR receptor in cancer cell progression, future studies could utilize a selective antagonist specific to the MOR receptor. This would greatly improve the specificity of our findings and provide direct information regarding the specific type of opioid receptors involved in cancer cell progression.

Another limitation of our study relates to the in vivo experiments focusing on MOR antagonists alone. The decision to focus the in vivo study on naloxone and its combination with cisplatin was based on the findings of our in vitro experiments. Furthermore, the effects of MOR agonists were already investigated by previous studies [30].

Our findings might provide a biologic explanation to previous epidemiologic studies on the relation between opioid use and cancer. Several studies reported that reduced doses of opioids following surgery for breast, lung, prostate, and colon cancers were associated with improved survival [25,26,27,28,29]. Furthermore, a recent randomized trial found that treatment with methylnaltrexone, a MOR antagonist, resulted in improved overall survival in patients with advanced cancer [54]. Since opioid use is prevalent amongst HNSCC patients [55,56], the effect on tumor progression and response to treatment is of utmost importance. Our study supports the hypothesis that opioids promote tumor progression while opioid antagonists might have a role in adjuvant therapy for HNSCC.

This preclinical study provides an important insight into the mechanism of MOR activation and inhibition in HNSCC cells. Upregulation and activation of MOR may promote tumor proliferation via the PI3K/Akt/mTOR pathway and blockage of MOR may sensitize HNSCC to chemotherapy. Since opioids are widely used in pain management of HNSCC patients, further understanding of the molecular events triggered by opioid agonists and antagonists is of paramount importance.

## Figures and Tables

**Figure 1 jcm-12-01277-f001:**
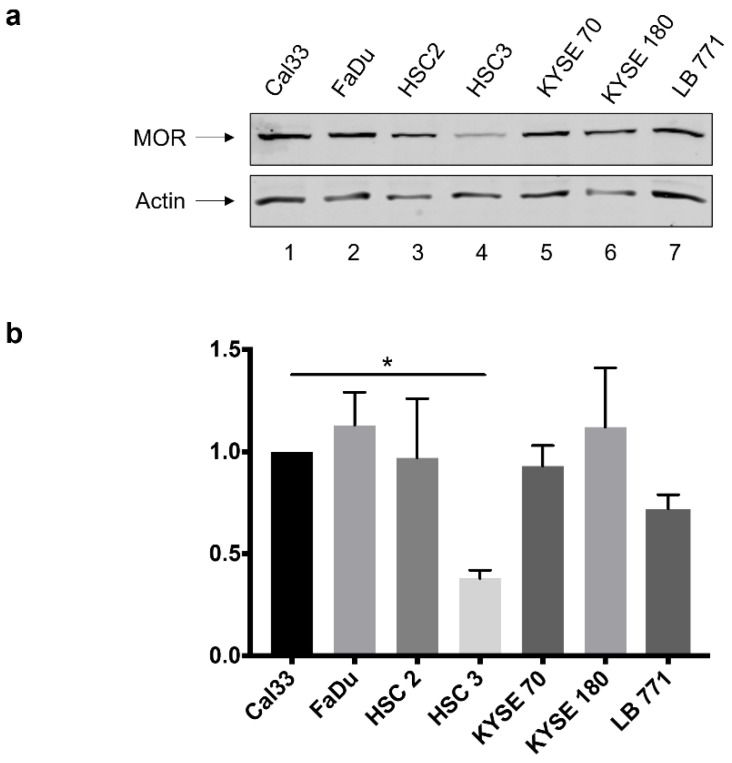
Western blot (WB) analysis for MOR-1 expression in different head and neck cancer cell lines. (**a**). WB for cell lines: Cal-33, FaDu, HSC2, HSC3, KYSE 70, KYSE 180, and LB771. All seven HNSCC cell lines demonstrate MOR-1 expression in Western blot. (**b**). Quantitative densitometry of WB MOR-1 expression while Cal-33 is the control, demonstrates a significant MOR-1 low expression in HSC-3 cell line. * *p*-value < 0.05.

**Figure 2 jcm-12-01277-f002:**
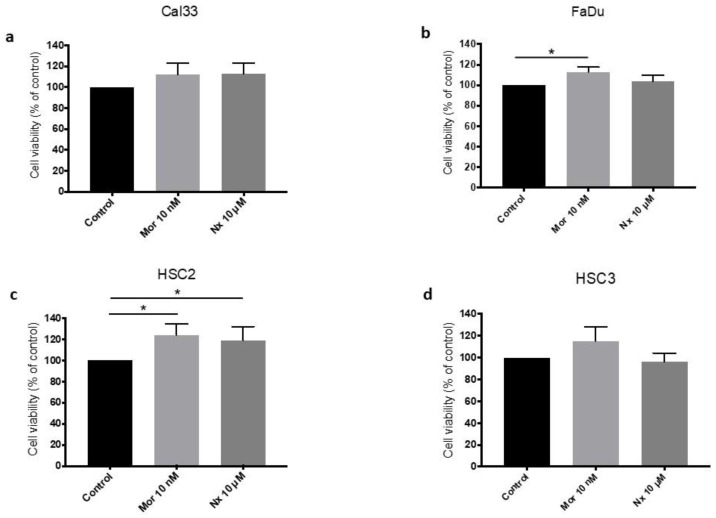
MOR agonist and antagonist effect on cell viability. Cell viability bar charts of Cal33 (**a**), FaDu (**b**), HSC2 (**c**), and HSC3 (**d**) cancer cell lines incubated without and with 10 nM morphine (Mor) or 10 µM naloxone (Nx), using XTT assay. Addition of morphine induces an increase in cell proliferation in all four HNSCC cell lines examined. * *p*-value < 0.05.

**Figure 3 jcm-12-01277-f003:**
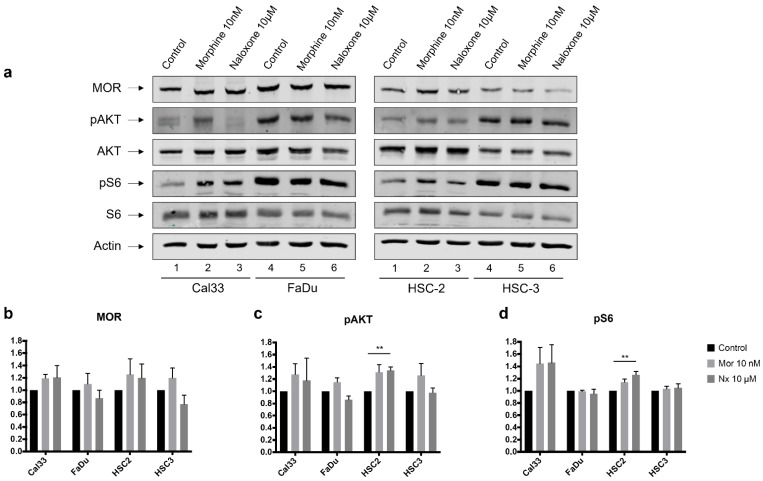
MOR agonist and antagonist effects on MOR-1 expression and AKT/PI3K signaling pathway. (**a**) Western blot analysis of MOR-1, phosphorylated or non-phosphorylated forms of AKT and S6 (proteins in the PI3K/AKT/mTOR axis) for four cancer cell lines (Cal-33, FaDu, HSC2, and HSC3), 24 h following incubation with: control medium, morphine (Mor) 10 nM, or naloxone (Nx) 10 µM. Quantitative densitometry of (**b**) MOR-1, (**c**) phosphorylated-AKT, or (**d**) phosphorylated-S6 expression with either morphine or naloxone relative to control in four cancer cell lines. Levels of phosphor-AKT and phosphor-S6 are altered when cells are treated with morphine or naloxone. ** *p*-value < 0.01.

**Figure 4 jcm-12-01277-f004:**
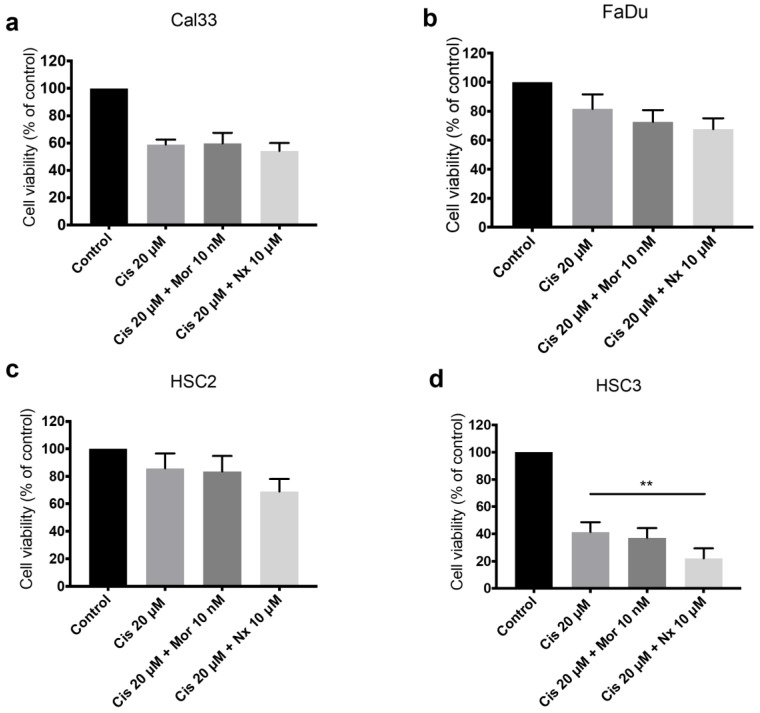
MOR agonist and antagonist effect on HNSCC cell response to cisplatin. XTT assay for cell viability of (**a**) Cal33, (**b**) FaDu, (**c**) HSC2, and (**d**) HSC3 cancer cell lines incubated with and without cisplatin (Cis) 20 µM, with addition of 10 nM (Mor) morphine or naloxone (Nx) 10 µM as combination treatment, using XTT assay. A significant decline by 46% for HSC-3 suggests a synergic cytotoxic effect between cisplatin and naloxone. ** *p*-value < 0.01.

**Figure 5 jcm-12-01277-f005:**
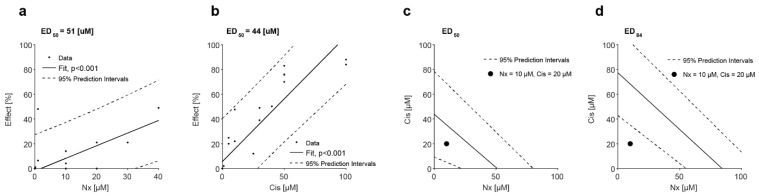
Isobolographic analysis of a synergistic effect between cisplatin (Cis) and naloxone (Nx). (**a**) Effect size as a function of Nx dose size. A linear model was fitted to the data yielding as significant fit (*p* < 0.001, F-test). ED50 = 51 [25, 80] (mean [95% confidence interval]). (**b**) Effect size as a function of Cis dose size. A linear model was fitted to the data yielding as significant fit (*p* < 0.001, F-test). ED50 = 44 [9.2, 78] (mean [95% confidence interval]). (**c**) Isobolographic analysis of Nx and Cis for effect level of 50%. A combined dose of Nx and Cis (10 µM and 20 µM, respectively) falls within the 95% confidence interval. (**d**) Isobolographic analysis of Nx and Cis for effect level of 84%. The combined dose falls well outside the 95% confidence interval, indicating a synergistic effect. MOR agonist and antagonist effects on cell migration.

**Figure 6 jcm-12-01277-f006:**
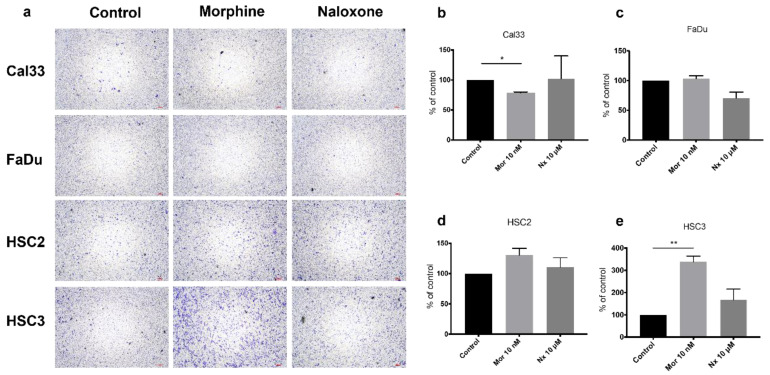
MOR agonist and antagonist effect on cell migration. (**a**). Cell migration assays for Cal-33, FaDu, HSC2, and HSC3. Cell lines were incubated with medium, 10 nM morphine, or naloxone 10 µM. Quantitative densitometry of cell migration for (**b**) Cal-33, (**c**) FaDu, (**d**) HSC2, and (**e**) HSC-3 with either morphine (Mor) or naloxone (Nx) relative to control. Morphine induces migration while naloxone inhibits it. * *p*-value < 0.05. ** *p*-value < 0.01.

**Figure 7 jcm-12-01277-f007:**
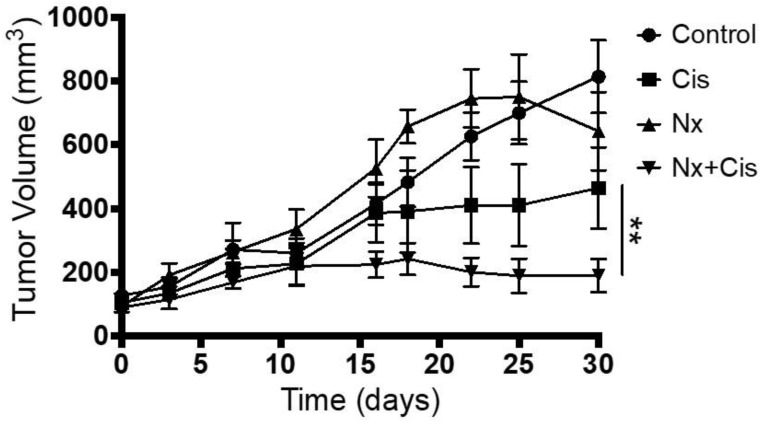
MOR antagonist effect on response to cisplatin treatment in HNSCC-bearing mice. In vivo intraperitoneal administration of naloxone with chemotherapy. HSC-3-tumor-bearing mice were administered with one of these four treatments: control, naloxone (Nx; 1 mg/kg), cisplatin (Cis; 3 mg/kg), and both naloxone and cisplatin. The addition of naloxone to cisplatin demonstrates a significant decrease in tumor volume compared to cisplatin alone. ** *p*-value < 0.01.

## Data Availability

The data presented in this study are available on request from the corresponding author. The data are not publicly available due to privacy.

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
