# Peer review of "Effect of Opioid Receptor Activation and Blockage on the Progression and Response to Treatment of Head and Neck Squamous Cell Carcinoma"

_jcm, 2023, doi:10.3390/jcm12041277_

Round 1

Reviewer 1 Report

The presented manuscript 'Effect of opioid receptor activation and blockage on the progression and response to treatment of HNSCC' by Levi and colleagues deals with a very interesting and clinically relevant issue. The in vitro and in vivo investigations were performed carefully with all required controls and data were presented clearly and understandable.

Data reveal that MOR (Mu Opioid Receptors) antagonists positively affect the response to Cisplatin treatment in HNSCC bearing mice. Would mTOR inhibitors such as Rapamycin show similar effect, since the authors demonstrated the relation between MOR activation and PI3K/AKT/mTOR signaling? Is there any literature in terms of combined treatment with Cisplatin and Rapamycin? This could be added to the discussion section.

Overall the manuscript is well written and supported by selected and current literature.

Author Response

Effect of opioid receptor activation and blockage on the progression and response to treatment of head and neck squamous cell carcinoma.

We would like to thank the Reviewers for their insightful and constructive criticism. We prepared our revised manuscript in light of their comments. In particular, we reframed the manuscript, added limitations, and clarified the presentation of the figures.   

Comment #1

Data reveal that MOR (Mu Opioid Receptors) antagonists positively affect the response to Cisplatin treatment in HNSCC bearing mice. Would mTOR inhibitors such as Rapamycin show similar effect, since the authors demonstrated the relation between MOR activation and PI3K/AKT/mTOR signaling? Is there any literature in terms of combined treatment with Cisplatin and Rapamycin? This could be added to the discussion section.

Addressing comment #1

We thank the reviewer for this important insight. Following this comment, a short review on mTOR inhibitors' effect was added to the discussion:

Discussion, pg. 10-11, lines 335-341: “Rapamycin, an mTOR inhibitor, induced apoptosis and inhibition of cell growth48. The synergism between mTOR inhibitors and Cisplatin was previously demonstrated on several different cancer cell types, including Head and neck cancer47,49–53. The pro-posed mechanism for the synergistic effects includes downstream regulatory mTOR mediated pathway, shifting the cell balance towards apoptosis and enhancing the effects of Cisplatin53. Our finding aligns with those previous studies and support the inhibitory effect of Naloxone via the PI3K/AKT/mTOR signaling pathway.

Reviewer 2 Report

The Authors stated that both ago-/and antagonists toward MOR will be determined, while in in vivo studies the experimental groups included only animals treated with an antagonist. Therefore, it would be nice to see the results also for agonist (here: morphine).

In line with this, on what basis did the Authorse choose to use these exact doses of drugs (in vivo experiments)? Was there any dose-response analysis? Please provide the EC50 value.

The Authors should provide with information regarding the way of drug administration while in vivo studies. This is missing in the methodology section.

Legend for figure 2 should be improved as there is no abbreviation for Nx. Similarly, figure 3.

The synergy between naloxone and cisplatin is rather a hypothesis. The Authors did not provide isobolographic analysis.

Also, if an opioid antagonist was administered together with cisplatin, the Authors should insert such information about simultaneous administration.

For figure 6, please provide the F values

Naloxone is a non-selective opioid antagonist, therefore also other opioidergic receptors (DOR, KOR) may be involved in the development of the cancer. An experiment with a selective antagonist at MOR would be sufficient, and these results will greatly imporved the paper. Also, a direct information will be given regarding the type of opioid receptors involved.

Round 2

Reviewer 2 Report

The Authors have greatly improved the manuscript. Therefore, in my opinion, the paper is suitable for publication